# Endurance Training Regulates Expression of Some Angiogenesis-Related Genes in Cardiac Tissue of Experimentally Induced Diabetic Rats

**DOI:** 10.3390/biom11040498

**Published:** 2021-03-25

**Authors:** Mojdeh Khajehlandi, Lotfali Bolboli, Marefat Siahkuhian, Mohammad Rami, Mohammadreza Tabandeh, Kayvan Khoramipour, Katsuhiko Suzuki

**Affiliations:** 1Department of Exercise Physiology, Faculty of Educational Sciences and Psychology, University of Mohaghegh Ardabili, Ardabil 5619913131, Iran; md.khajehlandi@uma.ac.ir (M.K.); m_siahkohian@uma.ac.ir (M.S.); 2Department of Sport Physiology, Faculty of Sport Sciences, Shahid Chamran University of Ahvaz, Ahvaz 6135783151, Iran; m.rami@scu.ac.ir; 3Department of Basic Sciences, Division of Biochemistry and Molecular Biology, Faculty of Veterinary Medicine, Shahid Chamran University of Ahvaz, Ahvaz 6135783151, Iran; m.tabandeh@scu.ac.ir; 4Department of Physiology and Pharmacology, Afzalipour Medical Faculty, Physiology Research Center and Neuroscience Research Center, Institute of Neuropharmacology, Kerman University of Medical Sciences, Kerman 7616913555, Iran; k.khoramipour@kmu.ac.ir; 5Faculty of Sport Sciences, Waseda University, Tokorozawa 359-1192, Saitama, Japan

**Keywords:** endurance training, angiogenesis, cardiac tissue

## Abstract

Exercise can ameliorate cardiovascular dysfunctions in the diabetes condition, but its precise molecular mechanisms have not been entirely understood. The aim of the present study was to determine the impact of endurance training on expression of angiogenesis-related genes in cardiac tissue of diabetic rats. Thirty adults male Wistar rats were randomly divided into three groups (N = 10) including diabetic training (DT), sedentary diabetes (SD), and sedentary healthy (SH), in which diabetes was induced by a single dose of streptozotocin (50 mg/kg). Endurance training (ET) with moderate-intensity was performed on a motorized treadmill for six weeks. Training duration and treadmill speed were increased during five weeks, but they were kept constant at the final week, and slope was zero at all stages. Real-time polymerase chain reaction (RT-PCR) analysis was used to measure the expression of myocyte enhancer factor-2C (MEF2C), histone deacetylase-4 (HDAC4) and Calmodulin-dependent protein kinase II (CaMKII) in cardiac tissues of the rats. Our results demonstrated that six weeks of ET increased gene expression of MEF2C significantly (*p* < 0.05), and caused a significant reduction in HDAC4 and CaMKII gene expression in the DT rats compared to the SD rats (*p* < 0.05). We concluded that moderate-intensity ET could play a critical role in ameliorating cardiovascular dysfunction in a diabetes condition by regulating the expression of some angiogenesis-related genes in cardiac tissues.

## 1. Introduction

Diabetes Mellitus (DM), with an estimated worldwide prevalence of 285 million patients, is an increasingly prevalent metabolic disorder [1] that is characterized by persistent hyperglycemia caused by insulin resistance or a lack of insulin [2,3]. It has been documented that patients with DM are at a higher risk of central and peripheral cardiovascular diseases, because DM harms structure and function of micro vessels [4,5]. Among the others, abnormal angiogenesis and collateral vessel formation are the most prevalent cardiovascular manifestations seen in DM patients [6,7].

Angiogenesis, as a multilevel and complex process of a new capillary formation, is controlled by angiostatic and angiogenic factors [8,9]. In the recent years, researchers found new players in the angiogenesis process that are regulated by a complex network of transcriptional factors including myocyte enhancer factor-2C (MEF2C), histone deacetylase-4 (HDAC4), and Calmodulin-dependent protein kinase II (CaMKII) [10]. MEF2C is a direct transcriptional target of endothelial transcription factors that play an important role in angiogenesis and vasculogenesis during vascular development [11]. Previous studies have revealed that MEF2C was significantly down-regulated in cardiac tissue of diabetic rats [12,13]. HDACs control biological processes by de-acetylation of histone and regulating accessibility of transcription factors to the gene promoter [14]. Among all HDACs, HDAC4 plays an important role in mediating cardiovascular diseases [15]. This transcription factor is phosphorylated and activated by CaMKII, a central culprit in the development of heart failure and cardiac arrhythmia [16], and negatively interacts with MEF2C to control its repressive activity [17]. The activities of HDAC4 and CaMKII (representatives of histone acetylation) are increased in the diabetic condition [18,19].

More recently, researchers have introduced testosterone as another influencing factor on angiogenesis mainly in cardiovascular system of diabetic patients [20]. Some studies have demonstrated that androgens paucity are common in men with diabetes, and testosterone deficiency may modulate endothelial angiogenesis [21,22]. The other component of diabetics is insulin resistance, which is also associated with an increased risk of the premature development of coronary artery disease. Therefore, it seems that the combination of testosterone deficiency and impaired glucose tolerance increases the risk of cardiovascular disease in patients with DM [21,22]. The recent studies have demonstrated that physiological testosterone therapy could improve insulin resistance [23].

Sedentary lifestyle and poor diet are common in patients with DM that worsen the patient’s condition [24]. Lifestyle modifiers as well as medication are considered as helpful approaches to tackle this problem. As a lifestyle modifier, exercise training could play an inevitable role in glycemic control [25,26,27,28]. Endurance training (ET), as the most popular type of exercise training, is considered as the most effective for DM patients [29,30]. There are also no reports on ET side effects in DM patients, highlighting its safety for DM treatment [27,28,31,32]. In addition, it has been suggested that ET could lead to vascular modifications associated with capillary density and angiogenesis [3,33,34,35] and result in cardiac remolding. For example, Ardakanizade et al. [36] examined the effects of long-term and mid-term ET on angiogenesis and reported higher gene expression of vascular endothelial growth factor B (VEGF-B), MEF2C, and matrix metalloproteinase-2 (MMP-2), and lower gene expression of HDAC4 and ANGPT-1 in the long- than mid-term ET.

Although exercise training and DM result in cardiac remodeling [34], the effect of moderate-intensity ET on expression of MEF2C, HDAC4, and CaMKII has not been entirely understood in DM. Therefore, our study was aimed to indicate whether a moderate-intensity ET can change the gene expression of MEF2C, HDAC4, and CaMKII in the cardiac tissue of diabetic rats.

## 2. Materials and Methods

### 2.1. Animal Models and Ethical Statement

The present study was in accordance with the guidelines for the care and use of laboratory animals approved by the Ethics Committee on the use of animal of Ardabil University of Medical Sciences (IR.ARUMS.REC.1398.251). In this experimental study, 30 male Wistar rats (249 ± 8.3 g) were obtained from the Iran Pasteur Institute (Tehran, Iran). Animals were kept under controlled in Plexiglas cages with a stable temperature of 23 ± 5 °C and humidity of 35 ± 5% on a cycle of 12-h light/dark. All animals had free access to standard food and water throughout the study with no difference in accessibility. Rats were randomly divided into three groups: Diabetic training (DT), sedentary diabetic (SD), and sedentary healthy (SH). Familiarization to treadmill was conducted for two weeks at the speed of ten meters per mins for 10–15 min, five days a week in all groups. At the end of familiarization and following an overnight fast, diabetes was induced to DT and SD groups by single intraperitoneal injection of Streptozotocin (STZ) at a dosage of 50 mg/kg (Sigma, St. Louis MO, USA). STZ was prepared in a fresh citrate buffer (0.5 M with pH 4.5), as described previously [37]. The same volume of citrate solution was injected into the SH group to simulate the stress of injection. Serum glucose level was measured 72 h after STZ injection using a portable glucometer (Roche Diagnostics K.K., Tokyo, Japan). Serum glucose higher than 250 mg/dL was considered as the benchmark to identify the diabetic rats [38], which was met by all rats in DT and SD groups. To eliminate the effects of food consumption, all blood samples were taken after 12-h of fasting. While the DT group rats conducted six-week ET, the SD and SH groups did not participate in exercise training program during the experiment period.

### 2.2. ET Protocol

Animals in the ET group performed an exercise protocol five days a week, for six weeks as shown in Table 1. Before and after each exercise training session, three-min warm-up and cool-down were carried out, respectively, and the treadmill slope was zero at all stages. Treadmill speed and training duration were kept constant during the final week (sixth week) to conserve the adaptations that resulted from 6 weeks ET [39]. All training sessions were conducted between 08:00–12:00 AM.

### 2.3. Biochemical Assays

Twenty-four hours after the last ET session, all animals were sacrificed by intraperitoneal injection of ketamine (75 mg/kg) and xylazine (5 mg/kg) following a 12-h fasting to measure testosterone concentration, and blood samples were taken from the animals’ heart and centrifuged for 15 min at 3000 rpm to obtain serum. Serums were kept at −20 °C until analysis. Testosterone concentration was measured using an ELISA kit (Monobind, Accubind, Costa Mesa, CA, USA) in a multiple ELISA reader (Bio Tek, Winooski, VT, USA) based on the recommended protocol by the manufacturer.

### 2.4. qRT-PCR Analysis 

Cardiac tissues were removed, submerged in liquid nitrogen, and kept at −70 °C until further analysis. The extraction of RNAs was performed by RNX^TM^ reagent according to the manufacturer’s procedure (Sina Clon Bioscience, Tehran, Iran). Concentration of RNA, and its purity were calculated by measuring the ratio of 260/280 nm optical density using Nanodrop spectrophotometry (Eppendorf, cologne, Germany), and values between 1.8–2 were defined as an acceptable purity. The cDNA synthesis was performed using qPCR^TM^ Green Master Kit for SYBR Green I^®^ (Yekta Tajhiz, Tehran, Iran) according to the instructions of the manufacturer. Real-time PCR was performed in Roche Light-Cycler detection system (Basel, Switzerland) with the following steps: Initial denaturation for 5 min at 95 °C and 45 cycles of denaturation for 15 s at 95 °C, annealing for 30 s at 60 °C, an extension for 20 s at 72 °C followed by melt curve analysis (50–99 °C) [40]. Glyceraldehyde-3-phosphate dehydrogenase (GAPDH) was used as the reference gene to measure relative gene expression. The results were evaluated by using 2^−ΔΔCt^ comparative method and Light Cycler SW1.1 software. The sequence of the primers is shown in Table 2.

### 2.5. Statistical Analysis

Data were analyzed using Statistical Package for Social Sciences (SPSS) version 23. Shapiro–Wilk normality test and one-way ANOVA were used to determine the normal distribution of variables and to compare changes between three groups, respectively. Tukey was also used as a post hoc test. The significance level was set at *p* < 0.05.

## 3. Results

Blood glucose and body weight changes are shown in Figure 1. Blood glucose levels increased significantly in the second week compared to before exercise in DT and SD groups, and this increase continued until the sixth week (*p* < 0.001). In addition, blood glucose levels were significantly reduced in the DT group compared to the SD group at the sixth week (*p* < 0.001). There was no significant change in blood glucose levels during the experimental period in the SH group (Figure 1A). STZ-treated animals (SD and DT groups) showed a decrease in body weight compared to the SH group in the fourth and sixth weeks (*p* < 0.005). DT group showed higher weights than SD in the fourth and sixth weeks, but this difference was not significant (Figure 1B).

As displayed in Figure 2, the SD group showed a significantly lower testosterone levels compared to the DT and SH groups (*p* < 0.001). In addition, the DT group showed significantly lower testosterone levels compared to the SH group (*p* < 0.001) (Figure 2).

The results of angiogenesis-related genes expression in cardiac tissues are shown in Figure 3A–C. MEF2C gene expression was higher in DT group compared to the SD group (*p* < 0.001), but it is lowered compared to the SH group significantly (*p* < 0.001) (Figure 3A). A significant difference in HDAC4 and CaMKII was observed in the DT group compared to the SD group (*p* < 0.001). Gene expression levels of HDAC4 and CaMKII were higher in both diabetic groups compared to the SH group (Figure 3B,C).

## 4. Discussion

This is the first study that examined the effect of moderate-intensity ET on MEF2C, HDAC4, and CaMKII gene expression and testosterone levels in diabetic hearts. It has been shown that exercise training improves glucose control and could affect both the structure and function of the myocardium, which could improve cardiovascular health in DM patients [41]. The results of the current study show significant weight loss in diabetic groups due to frequent urination, dehydration, and skeletal muscle atrophy. Researchers showed that exposure to high levels of glucose resulted in expression of muscle atrophy–related genes like Atrogin1 and Murf1 [20]. It should be noted that weight loss was lower in the DT group, because exercise training can stimulate muscle hypertrophy and inhibit muscle atrophy [42]. In the SH group, the weight gain process occurred naturally due to sufficient calorie intake. In addition, our results showed that ET controlled serum levels of glucose, increased MEF2C, and decreased HDAC4 and CaMKII gene expression and rose serum testosterone levels in diabetic rats. A study by Grossmann et al. [42] showed that testosterone levels are lower in STZ-induced diabetic rats compared to non-diabetes. Lower testosterone was also documented in diabetic men compared to non-diabetics [43,44]. Changes in the serum levels of testosterone could improve insulin sensitivity and oxidative capacity, as well as trigger anti-inflammatory processes and capillarization [45]. It has also been suggested that the regulatory effects of exercise on glucose metabolism and angiogenesis genes expression are facilitated by increased testosterone levels [46], which is in line with our results. Testosterone could probably increase expression of its target genes (Spred-1 and PI3KR2), which stimulate proliferation of vascular cells that are required for vessel angiogenesis [40]. In fact, testosterone is the principal masculine gonadal androgen hormone that modulates angiogenesis and endothelial functions [40]. While testosterone deficiency is predominant in STZ-induced diabetic rats, it seems that exercise constitutes this deficiency, leading to increased angiogenesis genes expression.

To elucidate the effect of ET on the cardiac angiogenesis process, we studied gene expression of MEF2C, HDAC4, and CaMKII in the diabetic heart, and our results showed a higher gene expression of MEF2C and lower gene expression of HDAC4 and CaMKII in DT than another diabetic group. It seems that down-regulation and deacetylation of HDAC4 allowing MEF2C to activate angiogenesis process [47,48]. We believe that lower expression of HDAC4 in the DT group is accompanied by higher expression of the MEF2C. These results are in line with the previous research findings on the role of HDAC4 in the angiogenesis process [47,48]. On the other hand, ET induced an increase in the antioxidant potential, which could be another explanation for changes in MEF2C and HDAC4 gene expression because it has been reported that oxidative stress, as a novel phosphorylation-independent post-translational modification, regulates subcellular localization of MEF2C in cardiomyocytes [49]. It has been indicated that increased MEF2C gene expression can up-regulate vascular endothelial growth factor (VEGF)-B, which is a key regulator of angiogenesis [11]. Although the amount of VEGF-B has not been measured in the current study, due to financial limitations, based on the previous studies [36], we hypothesize that ET could increase the gene expression of VEGF-B. It is observed that HDAC4 down-regulation increases angiogenesis through stimulation of VEGF-B gene expression [50], and it has also been reported in the cerebral ischemia that higher expression of the HIF-VEGF signaling gene has be seen through the phosphorylation of the HDAC4 protein [51].

Another finding of the present study was down-regulation of CaMKII gene expression by ET, which is in accordance with other studies [52,53]. Stolen et al. [53] has shown that in diabetic rats with reduced cardiomyocyte contractile function, Ca^2+^ handling and chronically increased cardiac CaMKII aerobic interval exercise training resulted in reduced CaMKII levels. It has been demonstrated that CaMKII is capable of regulating the angiogenesis factors [19], and stimulates glucose uptake, sarcolemma ion fluxes, energy production, sarcoplasmic reticulum Ca^2+^ release/reuptake, and myocyte contraction/relaxation during acute activation [54]. Contrary to physiological condition, in a disease condition such as diabetes, CaMKII leads to mitochondrial dysfunction, cell fibrosis, remodeling of ion channel, inefficient substrate utilization, impaired intracellular Ca^2+^ handling, inflammation, and a contractile dysfunction leading to increased risk of arrhythmias [55]. CaMKII activity was up-regulated in the heart of diabetic rats in our study. CaMKII modulates transcription of HDAC4 at multiple levels [56], as we see higher HDAC4 gene expression in diabetic rats but both of them were controlled by ET. Therefore, ET-induced decrease in CaMKII gene expression may leads to decrease of HDAC4 as well. It seems that inhibition of CaMKII expression could possibly be a therapeutic strategy for DM by remodeling and promote angiogenesis in cardiac tissue. Beyond, it is possible that a decrease in CaMKII gene expression attenuates HDAC4 gene expression [56], but paradoxically lead to a greater increase in MEF2C. Thus, we provide support for the hypothesis that MEF2C regulation is under the control of HDAC4 and CaMKII during the regulatory adaptation to moderate-intensity ET in diabetic myocardium. However, more studies is needed to prove these results and shed light on the exact it’s signaling pathways.

## 5. Conclusions

Taken together, the results of this study indicated that six weeks of moderate-intensity ET allowed more effective control of glucose homeostasis, increased testosterone levels, and induced up-regulation of MEF2C and down-regulation of HDAC4 and CaMKII in cardiac tissue of diabetic rats. These results suggest improvements in managing the diabetic-induced cardiac dysfunction. However, future studies should cover our limitation by analyzing angiogenesis markers as well.

## Figures and Tables

**Figure 1 biomolecules-11-00498-f001:**
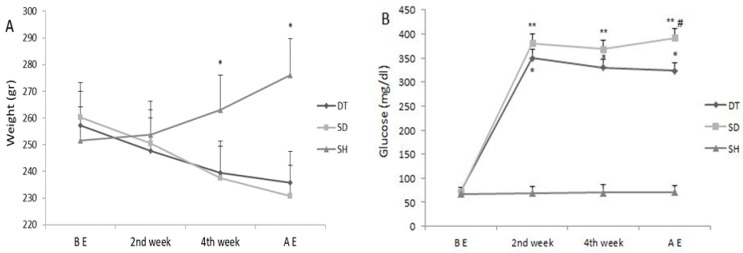
(**A**) Weight change during and after ET. *; Significant difference between the SH group with DT and SD groups in 4th week and AE. (**B**) Changes in the serum levels of glucose during six weeks ET period. *; significant difference (*p* < 0.001) between DT and SH groups in 2nd week, 4th week, and AE with SH group.**; significant difference (*p* < 0.001) between SD and SH groups in 2nd week, 4th week, and AE with SH group **^#^**; significant difference (*p* < 0.001) between DT and SD groups AE. Abbreviations; ET: Endurance training, BE: Before Exercise, AE: After exercise, DT: Diabetic training, SD: Sedentary diabetic, SH: Sedentary healthy. Data are expressed as mean ± SEM.

**Figure 2 biomolecules-11-00498-f002:**
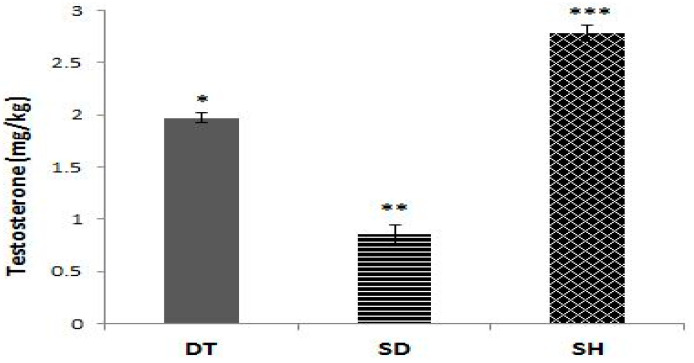
Different serum levels of testosterone in the DT, SD, and SH groups after six weeks of ET. *; significant difference (*p* < 0.001) between DT and SD groups, **; significant difference (*p* < 0.001) between SD and SH groups, ***; significant difference (*p* < 0.001) between DT and SH groups. Abbreviations; DT: Diabetic training, SD: Sedentary diabetic, SH: Sedentary healthy, ET: Endurance training. Data are expressed as mean ± SEM.

**Figure 3 biomolecules-11-00498-f003:**
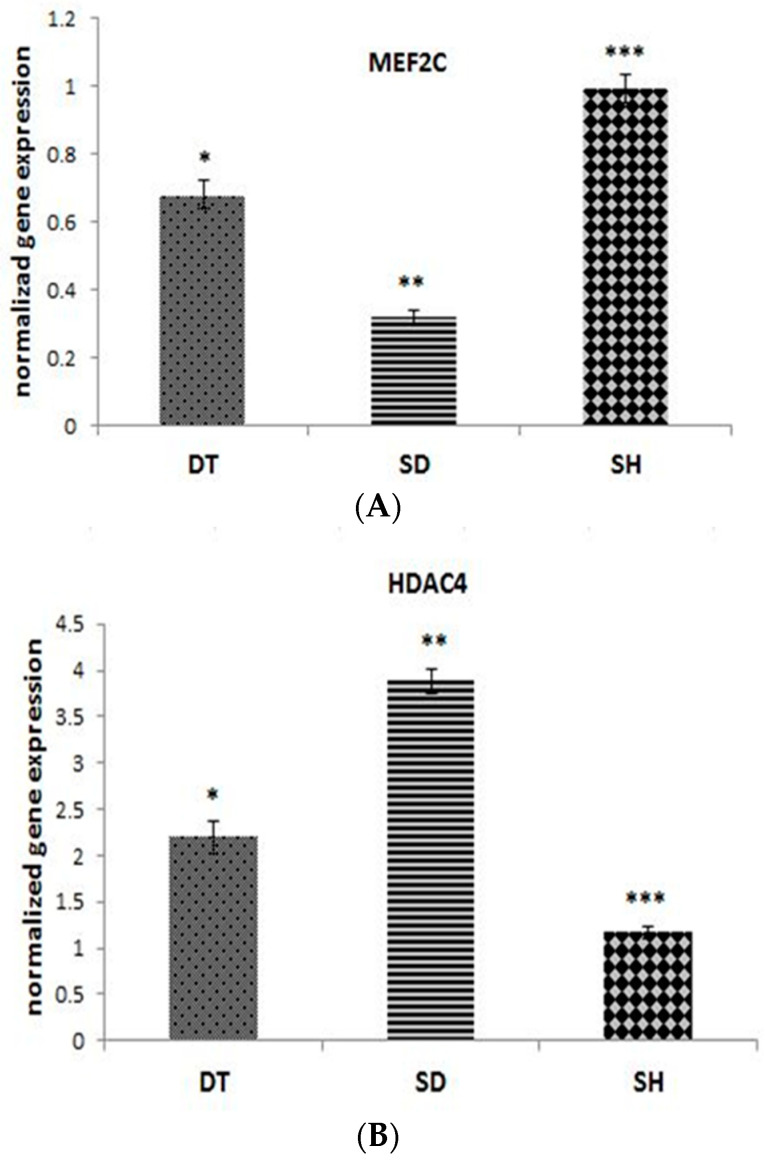
The normalized gene expression of MEF2C (**A**), HDAC4 (**B**), and CaMKII (**C**) at the end of six weeks ET in DT, SD, and SH groups. *; significant difference *p* < 0.001 between DT and SD groups, **; significant difference (*p* < 0.001) between SD and SH groups, ***; significant difference (*p* < 0.001) between DT and SH groups. Abbreviations; ET: Endurance training, DT: Diabetic training, SD: Sedentary diabetic, SH: Sedentary healthy. Data are expressed as mean ± SEM.

**Table 1 biomolecules-11-00498-t001:** Endurance training protocol in different weeks.

Weeks	First Week	Second Week	Third Week	Forth Week	Fifth Week	Sixth Week
**Duration (minutes)**	10	20	20	30	30	30
**Speed (m/minutes)** **Slope**	10 0	10 0	15 0	15 0	17–18 0	17–18 0

**Table 2 biomolecules-11-00498-t002:** The sequence of primers for quantitative real-time polymerase chain reaction (RT-PCR).

Genes	Forward	Reverse	Amplicon Size (bp)
**GAPDH**	AGTTCAACGGCACAGTCAAG	TACTCAGCACCAGCATCACC	119
**MEF-2C**	CTTCAACAGCACCAACAAGC	TCAATGCCTCCACAATGTCC	125
**HDAC4**	CTCTGCCAAATGTTTCGGGT	CAAGCTCATTTCCCAGCAGA	149
**CaMKII**	AGTGACACCTGAAGCCAAAG	GTCAAGATGGCACCCTTCAA	198

## Data Availability

All data related to this study is available at https://uma.ac.ir/index.php?slc_lang=en.

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
