# Peer review of "Endurance Training Regulates Expression of Some Angiogenesis-Related Genes in Cardiac Tissue of Experimentally Induced Diabetic Rats"

_biomolecules, 2021, doi:10.3390/biom11040498_

Round 1

Reviewer 1 Report

Diabetes mellitus (DM) is one of the major risk factors for cardiovascular disease, leading to impair angiogenesis. Exercise is a therapeutic strategy to improve overall cardiovascular health during diabetes. Khajehlandi et al., found that exercise increased MEF2C and decreased HDAC4 and CaMKII gene expression levels in diabetic rats. Reviewer has following concerns in this manuscript.

Authors used STZ (50 mg/kg BW, IP) to induce diabetes by citing previous study (Adenowo et al., TJPR 2014 (37)). However, this previous article used alloxan-induced DM (not STZ). How authors determine dose of STZ and mode of injection.

Many studies used 40-60 mg/Kg STZ and injected intravenously. But authors used intraperitoneal (IP) injection of single dose of STZ. Authors need to convince method of DM induction.

Authors need to discuss about the reason for body weight change in DM+Sed and DM+Exe mice.

The outcome of this study is that exercise increased angiogenesis in DM mice. Thus author needs to show functional significance of this study by showing angiogenesis marker (CD31 or other) staining in cardiac tissue.

Also authors also needs to show VEGF-B mRNA expression.

Author Response

Dear reviewer,

Thank you for your helpful comments. We tried applying all of them. In addition, we tried to fine tone the English language as well. Please see our replies below. Please let us know if you have more concerns.

Authors used STZ (50 mg/kg BW, IP) to induce diabetes by citing previous study (Adenowo et al., TJPR 2014 (37)). However, this previous article used alloxan-induced DM (not STZ). How authors determine dose of STZ and mode of injection.

It was a writing error and the reference has been revised. The method we used was described in details in the following studies. In short, we used STZ at a dose of 50 mg/kg for intraperitoneal injection.

Kurd, M., Valipour, V.,  Tavakoli, S A., Gahreman D E., Effects of endurance training on hippocampus DJ-1, cannabinoid receptor type 2 and blood glucose concentration in diabetic rats. J Diabetes Investig Vol. 10 No. 1 January 2019.

Ne’eman, Z., et al. Localization of glycogen in the placenta of diabetic rats: a light and electron microscopic study. Placenta 1987; 8: 201–208.

Keshvari M., Rahmati M., Mirnasouri R., Chehelcheraghi F. Effects of endurance exercise and Urtica dioica on the functional, histological and molecular aspects of the hippocampus in STZ-Induced diabetic rats. Journal of Ethnopharmacology. 2020. Volume 256. https://doi.org/10.1016/j.jep.2020.112801.  

Many studies used 40-60 mg/Kg STZ and injected intravenously. But authors used intraperitoneal (IP) injection of single dose of STZ. Authors need to convince method of DM induction.

In many studies on diabetes and its pathological effects, high doses of STZ (e.g 60 mg/kg has been used) intraperitoneally such as the study by Fang Meng et al., (please see it below) because they needed the rats to be alive for up to seven days however we needed our rates to be alive lone enough to finish 6 weeks of training and so used the lowest dose that can induce Diabetes  (50 mg/kg). Another reason for using low dose is our university guideline “use the minimum dose of drag to induce disease”. As in our previous studies 50 mg/kg STZ resulted in Diabetes, we chose such dosage for this study as well.

Followings are four studies, which used different dosage of STZ, but all injected intraperitoneally.

60 mg/Kg STZ= Meng, X.-F., Wang, X.-L., Tian, X.-J., Yang, Z.-H., Chu, G.-P., Zhang, J., . . . Zhang, C. (2014). Nod-like receptor protein 1 inflammasome mediates neuron injury under high glucose. Molecular neurobiology, 49(2), 673-684.

45 mg/Kg STZ=Keshvari M., Rahmati M., Mirnasouri R., Chehelcheraghi F. Effects of endurance exercise and Urtica dioica on the functional, histological and molecular aspects of the hippocampus in STZ-Induced diabetic rats. Journal of Ethnopharmacology. 2020. Volume 256. https://doi.org/10.1016/j.jep.2020.112801

45 mg/Kg STZ= Rahmati M, Gharakhanlou R, Movahedin M, Mowla SJ, Khazani A, Fouladvand M, Jahani Golbar S. Treadmill Training Modifies KIF5B Motor Protein in the STZ-induced Diabetic Rat Spinal Cord and Sciatic Nerve.Arch Iran Med. 2015; 18(2):94 – 101.

50 mg/Kg STZ= Kurd, M., Valipour, V.,  Tavakoli, S A., Gahreman D E., Effects of endurance training on hippocampus DJ-1, cannabinoid receptor type 2 and blood glucose concentration in diabetic rats. J Diabetes Investig Vol. 10 No. 1 January 2019

Authors need to discuss about the reason for body weight change in DM+Sed and DM+Exe mice.

This explanation was added to the text.

“The results of the current study show significant weight loss in diabetic groups due to frequent urination, dehydration and skeletal muscle atrophy. Researches showed that exposure to high levels of glucose result in expression of muscle atrophy–related genes like Atrogin1 and Murf1 [20].  However, this weight loss was lower in the DT group, because exercise training can stimulate muscle hypertrophy and inhibit muscle atrophy [42].”

Reviewer 2 Report

In this paper the author demonstrated MEF2C , HDAC4 and CaMKII genes and testosterone  modulations by exercise. This result is in agreement with some other papers more particularly with skeletal muscle. However this paper seems original in the heart. Since this molecules are also key actors of metabolism confirmation of the hypothesis of angiogenesis modulation would enhance the message. Per se experiments of this paper are well done and convincing. The paper need to be red careffully

Minor concerns

Typo error in line 85.”MD” is probably DM

Line 127 refer to the use of STZ at 50mg/kg while in abstract it is indicated 30mg/kg

Line 128 Citrate buffer in STZ solution appears to high probably wrong

Line 128 ref 37 doesn’t describe STZ preparation no more buffer

Line 177 “glucose levels 177 increased continuously in the DT and SD groups (P˂0.001),” This is not really what is shown in the figure 1. Glucose concentration plateau after 2weeks and remained higher in SD group

Line 227 “however, this weight loss was 227 lower in the DT group in diabetic rats [20]” not find in your results

Line 261 “it has been reported 259 that oxidative stress, as a novel phosphorylation-independent post-translational modifi-260 cation, regulate subcellular localization of MEF2C in cardiomyocytes [49]” Regulates

Line 291 there is no proof of angiogenesis only expression of genes

Line 294 “Thus, we pro-294 vide support for the hypothesis that MEF2C regulation is under the control of HDAC4 295 and CaMKII during the regulatory response adptation to moderate-intensity ET in dia-296 betic myocardium.” The study is accordance with that hypothesis bus doesn’t support it because no proof was brought but only association. Typo error  “adaptation”

Author Response

Dear reviewer,

Thank you for your helpful comments. We tried applying all of them. Please see our replies bellow.

Please let us if you have more concerns.

Minor concerns

Typo error in line 85.”MD” is probably DM

Thank you. We have revised it to “DM”

Line 127 refer to the use of STZ at 50mg/kg while in abstract it is indicated 30mg/kg

50 mg/kg is correct. The abstract has been revised.

Line 128 Citrate buffer in STZ solution appears to high probably wrong

To the best of our knowledge, citrate is used as STZ buffer. Here are two examples of the studies that used citrate.

Kurd, M., Valipour, V.,  Tavakoli, S A., Gahreman D E., Effects of endurance training on hippocampus DJ-1, cannabinoid receptor type 2 and blood glucose concentration in diabetic rats. J Diabetes Investig Vol. 10 No. 1 January 2019

Ne’eman, Z., et al. Localization of glycogen in the placenta of diabetic rats: a light and electron microscopic study. Placenta 1987; 8: 201–208

Line 128 ref 37 doesn’t describe STZ preparation no more buffer

Sorry for this writing error. The reference has been revised.

Line 177 “glucose levels 177 increased continuously in the DT and SD groups (P˂0.001),” This is not really what is shown in the figure 1. Glucose concentration plateau after 2weeks and remained higher in SD group

Yes, this is completely right. It has been revised.

Blood glucose levels increased significantly in the second week compared to before exercise in DT and SD groups and this increase continued until the sixth week (P˂0.001). In addition, blood glucose levels were significantly reduced in the DT group compared to the SD group at the sixth week (P˂0.001). There was no significant change in blood glucose levels during the experimental period in the SH group (Figure 1, A).

Line 227 “however, this weight loss was lower in the DT group in diabetic rats [20]” not find in your results.

This sentence has been revised.

“STZ-treated animals (SD and DT groups) showed decrease in body weight compared to the SH group in the fourth and sixth weeks (P<0.005). While DT group showed higher weigh than SD in weeks 4th and 6th but this difference was not significant”

Line 261 “it has been reported that oxidative stress, as a novel phosphorylation-independent post-translational modifi-cation, regulate subcellular localization of MEF2C in cardiomyocytes [49]” Regulates

It has been revised.

Line 291 there is no proof of angiogenesis only expression of genes

Thank you for this helpful comment. We add the following sentences to answer your concern.

“It has been indicated that increase MEF2C gene expression can up-regulate vascular endothelial growth factor (VEGF)-B gene expression that is a key regulator of angiogenesis [11]. Although the amount of VEGF-B has not been measured in the current study, due to financial limitation, based on the previous studies [36], we hypothesizes that ET could increase the gene expression of VEGF-B”

Line 294 “Thus, we provide support for the hypothesis that MEF2C regulation is under the control of HDAC4 and CaMKII during the regulatory response adptation to moderate-intensity ET in diabetic myocardium.” The study is accordance with that hypothesis bus doesn’t support it because no proof was brought but only association. Typo error  “adaptation”

We added this sentence to show the uncertainty.

“However, more studies is needed to prove these results and shed light on the exact signaling pathways”

In addition, the following sentence we added to the end of conclusion.

“However, future studies should cover our limitation by analyzing angiogenesis markers as well”

“Adaptation” has been revised.

Round 2

Reviewer 1 Report

Authors addressed reviewer's concern. However, adding VEGF mRNA expression and angiogenesis data in cardiac tissue will improve the paper quality as mentioned in previous comments.

This manuscript is a resubmission of an earlier submission. The following is a list of the peer review reports and author responses from that submission.

Round 1

Reviewer 1 Report

Dear Authors,

the study shows that a moderate-intensity endurance training improves glucose control, increases testosterone levels, induces up-regulation of MEF2C and down-regulation of HDAC4 and CaMKII in cardiac tissue of diabetic rats. Although the aim of this study is interesting and original, however the research design is not appropriate. The methods show some significant limitations as the absence of: 1. data concerning the expression of the examined genes in the 3 groups (DT, SD and SH) at the baseline before 6 weeks of endurance training; 2. the statistical analysis between pre- and post-intervention data within each group and among the three groups in the pre-intervention condition; 3. the measurement of a marker that indicates the efficiency of the endurance training protocol (skeletal muscle hypertrophy, capillarization...); 4. random assignment of animals into the three groups. In addition, the article needs an extensive editing of English language.

Reviewer 2 Report

Summary:  The authors wanted to investigate potential mechanisms for the benefits of endurance training on diabetes-associated cardiovascular dysfunction.  Utilizing RT-PCR to measure gene expression in cardiac tissue of sedentary health (SH), sedentary diabetes (SD; streptozotocin-induced diabetes), and diabetic trained (DT) mice, they concluded that endurance training may benefit the diabetes-induced cardiac dysfunction by regulation the expression of certain angiogenesis-related factors in the heart.

Comments:

  1. Please state if the mice were fasted prior to sacrifice, and if that had any implications on glucose levels.
  2. The weight of the diabetic mice were lower than control mice according to Fig 1A.  Was food intake different between these groups?  Please comment on the weight difference.
  3. The main and novel finding of these study is displayed in figure 3.  However there is significant literature suggesting that gene expression in cardiac tissue does no routinely translate to differences in protein levels.  Are the authors able to analyze protein levels, as this would considerably strengthen their findings.
  4. Is SD actually being reported in the error bars or is that standard error?
  5. Please correct numerous grammatical and typographical errors throughout manuscript.  (ex, proccess and reserchers) on line 52.
  6. Omit text starting on line 94 which are the instructions for the author.